# *i*LRM: An Iterative Large 3D Reconstruction Model

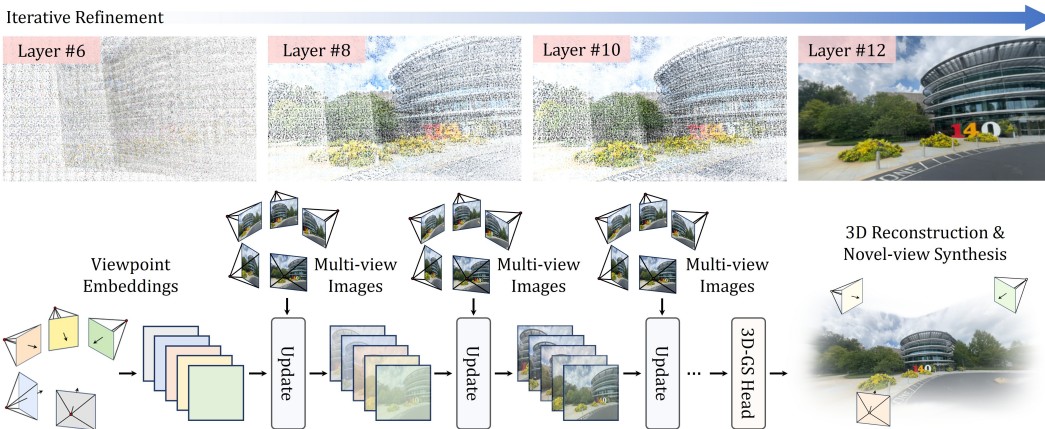

Figure 1: The overall architecture and qualitative results of the proposed *iLRM*.

## Abstract

Feed-forward 3D modeling has emerged as a promising approach for rapid and high-quality 3D reconstruction. In particular, directly generating explicit 3D representations, such as 3D Gaussian splatting, has attracted significant attention due to its fast and high-quality rendering, as well as numerous applications. However, many state-of-the-art methods, primarily based on transformer architectures, suffer from severe scalability issues because they rely on full attention across image tokens from multiple input views, resulting in prohibitive computational costs as the number of views or image resolution increases. Toward a scalable and efficient feed-forward 3D reconstruction, we introduce an iterative Large 3D Reconstruction Model (*iLRM*) that generates 3D Gaussian representations through an iterative refinement mechanism, guided by three core principles: (1) decoupling the scene representation from input-view images to enable *compact 3D representations*; (2) decomposing fully-attentional multi-view interactions into a *two-stage attention* scheme to reduce computational costs; and (3) injecting *high-resolution information at every layer* to achieve high-fidelity reconstruction. Experimental results on widely used datasets, such as RE10K and DL3DV, demonstrate that *iLRM* outperforms existing methods in both reconstruction quality and speed.

## 1 Introduction

Since the recent success of 3D Gaussian Splatting (3D-GS) (Kerbl et al., 2023), significant progress has been made in leveraging this 3D representation for building generalizable feed-forward 3D reconstruction models (Charatan et al., 2024; Tang et al., 2025; Xu et al., 2024b; Chen et al., 2025; 2024d; Xu et al., 2025; Zhang et al., 2025). These methods typically train large neural networks to transform multi-view input images into feature representations, then regress Gaussian attributes. In contrast to per-scene 3D-GS optimization approaches (Kerbl et al., 2023; Mallick et al., 2024; Lu et al., 2024; Fang & Wang, 2024), these feed-forward models can reconstruct 3D scenes in a single forward pass, offering near real-time performance. Moreover, the prior knowledge learned from large-scale datasets (Zhou et al., 2018; Ling et al., 2024; Deitke et al., 2023; 2024) allows them to effectively

generalize to unseen scenes. While their reconstruction quality often lags behind that of per-scene optimization methods, fast reconstruction speed and generalization capability mark a promising step toward the long-standing goal of achieving accurate and real-time 3D scene reconstruction.

Among the promising approaches, pixel-aligned Gaussian models (Charatan et al., 2024; Szymanowicz et al., 2024b; Zheng et al., 2024) have emerged as the de facto standard, leveraging decades of advances in network architectures developed for numerous image-based tasks. While these models have proven effective, they also exhibit certain limitations. In particular, since they generate per-pixel Gaussians directly from the input images, the image resolution determines the number of Gaussians produced, which can lead to an excessive number of redundant Gaussians. For example, given input images at 1K resolution across 200 viewpoints (a scale comparable to the bicycle scene in the mip-NeRF 360 dataset (Barron et al., 2022)), these methods would produce 200 million Gaussians, despite prior studies (Lee et al., 2024; Fan et al., 2024; Chen et al., 2024c; Lee et al., 2025) demonstrating that such scenes can be efficiently represented with around 0.5 million Gaussians. To mitigate this issue, several techniques have been proposed, such as Gaussian regularization (Ziwen et al., 2025) and feature fusion (Wang et al., 2025b). Alternatively, the network architecture can also be designed to generate fewer Gaussians, for example by downsampling the output resolutions. However, these strategies still require processing high-resolution multi-view images and therefore do not address another fundamental limitation of these models: high computational and memory demands.

A significant portion of computational and memory overhead arises from modeling interactions across multiple input views in feed-forward 3D reconstruction models. For instance, GS-LRM (Zhang et al., 2025) performs full attention over all image tokens from every input view, leading to a quadratic increase in complexity with respect to both the number of views and image resolution. MVSplat (Chen et al., 2025) and DepthSplat (Xu et al., 2025) construct and process dense cost volumes for each view, further contributing to the overall computational demands. While one might attempt to alleviate this burden by reducing the input image resolution or using a sparser set of views, such strategies risk discarding essential geometric and appearance information required for accurate reconstruction.

Beyond the computational complexity and the inefficiency of the generated representations, we also question whether the prevailing formulation, casting 3D reconstruction as a sequence-to-sequence problem that maps entire sets of image tokens to dense, pixel-aligned Gaussians, is fundamentally well-suited to the nature of the task. While this formulation has achieved impressive results (even without explicit 3D inductive biases (Zhang et al., 2025; Ye et al., 2024)), it remains primarily a one-shot 3D scene generation process. In contrast, the recent optimization-based methods (Kerbl et al., 2023; Mallick et al., 2024) follow a fundamentally different strategy: they treat reconstruction as an iterative refinement process, where each iteration involves rendering the current scene estimate, measuring reconstruction error, and updating the representation accordingly. This loop enables the model to progressively capture finer geometric and appearance details while ensuring strong 3D consistency. The success of these methods suggests that high-quality 3D reconstruction may benefit not only from expressive representations but also from feedback-driven iterative refinement, a trait largely absent in existing feed-forward 3D models.

In this paper, we introduce *iLRM*, an iterative large 3D reconstruction model that effectively 1) incorporates the principles of feedback-driven refinement, while also 2) addressing the computational burden and representational inefficiencies inherent in existing feed-forward approaches. As illustrated in Fig. 1, the network (acting as an optimizer) transforms the embedding features (analogous to updating the 3D-GS representation) at each layer (analogous to each optimization step), based on multi-view image tokens (serving as gradient-like signals). This design allows the model to iteratively update the scene representation at every layer based on feedback from the input images, effectively mimicking the optimization process within a feed-forward architecture. Through this process, the learned neural network jointly examines the input view images and the evolving scene representation to identify where and how to make targeted updates that improve reconstruction quality.

Another core design principle of our approach is to decouple the representation, later transformed into 3D Gaussians, from direct dependence on input images, addressing the computational complexity and redundancy that arise in architectures that generate pixel-aligned Gaussians directly from multi-view inputs. By decoupling the representation and the input images, we can use low-resolution representations to produce a compact set of Gaussians while still leveraging high-resolution input images for detailed guidance.

In addition, we propose an efficient mechanism for modeling the interaction between the representations and the input images. A naïve approach would involve computing full attention between all tokens across views, which quickly becomes computationally prohibitive. To overcome this, we initialize our representation using viewpoint embeddings, each tied to a specific input view. Interaction modeling is then split into two stages. First, we perform cross-attention between each viewpoint embedding and its corresponding image, which is highly efficient due to the one-to-one mapping. Next, we apply self-attention across all viewpoint embeddings. Importantly, since this second stage operates over a low-resolution representation space, it remains computationally tractable while facilitating global information exchange across views. Overall, this scalable design significantly reduces computational and memory overhead and allows for the incorporation of more viewpoints, thereby improving reconstruction fidelity.

We have comprehensively evaluated the proposed method on the large-scale datasets, RealEstate10K (Zhou et al., 2018) and DL3DV (Ling et al., 2024). The experimental results demonstrate that *iLRM* achieves superior rendering quality while substantially reducing both computational and memory overhead compared to recently proposed feed-forward Gaussian models. Moreover, in wide-coverage settings, our method outperforms optimization-based 3D-GS approaches, requiring only 0.5 seconds for inference while achieving higher fidelity reconstructions.

## 2 METHOD

### 2.1 MOTIVATION AND PROBLEM STATEMENT

Existing generalizable 3D Gaussian reconstruction methods process multi-view images in an end-to-end fashion, often employing techniques such as epipolar line sampling (Charatan et al., 2024), plane-sweep stereo (Chen et al., 2025; 2024d; Xu et al., 2025), or full-resolution attention (Zhang et al., 2025; Xu et al., 2024b; Ziwen et al., 2025) to enforce multi-view consistency. While effective, these strategies introduce significant computational and memory overhead, limiting their scalability.

To address these challenges, we propose *iLRM*, a novel feed-forward 3D reconstruction framework that decouples Gaussian generation from direct dependence on input images. Instead of generating pixel-aligned Gaussians, *iLRM* initializes viewpoint-centric embeddings as the basis for constructing the 3D scene. These embeddings are then iteratively refined via cross-attention with multi-view image features, enabling the model to efficiently fuse geometric and appearance cues across views.

We start with $N$ multi-view images $\{I_i\}_{i=1}^N$ and corresponding camera poses $\{C_i\}_{i=1}^N$. Based on this setup, our goal is to train a model $f_\theta$ that maps a set of viewpoints to 3D Gaussians, leveraging the associated multi-view images as visual cues throughout the reconstruction pipeline. More formally,

$$f_\theta : \{(C_i, I_i)\}_{i=1}^N \mapsto \{(\mu_k, \alpha_k, \Sigma_k, c_k)\}_{k=1}^{H^v W^v N}, \tag{1}$$

where $f_\theta$ is modeled as a feed-forward network with the model parameter $\theta$. $\mu_k, \alpha_k, \Sigma_k, c_k$ are attributes of 3D Gaussians, representing the mean, opacity, covariance, and color, respectively, while $H^v$ and $W^v$ denote the height and width of the generated Gaussians for each camera viewpoint. It is important to note that they do not correspond to the resolution of the input images. Following prior works, we train our model using held-out target images along with their corresponding camera poses, enabling high-quality novel view synthesis.

### 2.2 ARCHITECTURAL DESIGN

We propose an end-to-end transformer that directly regresses 3D Gaussian parameters from viewpoint embeddings. To compensate for the absence of direct image input, we enrich these embeddings at each layer via cross-attention with multi-view image features. The resulting embeddings are further refined through self-attention to capture global dependencies across viewpoints.

**Viewpoint tokenization.** Following previous works (Tang et al., 2025; Zhang et al., 2025; Jin et al., 2024), we employ a Plücker ray embedding for each input view using the camera poses. Specifically, given the intrinsic, rotation, and translation, we construct the Plücker ray embeddings for each viewpoint. We then divide these viewpoint embeddings into non-overlapping patches of size $p \times p$, and reshape each patch into a 1D vector, resulting in a tensor of shape $H^v W^v / p^2 \times 6p^2$. Then, we encode it using a single linear layer to produce viewpoint tokens, $V_i^{(0)} \in \mathbb{R}^{H^v W^v / p^2 \times d}$. Plücker coordinates inherently capture spatial variations across pixels and views, allowing them to effectively differentiate between patches. As a result, we do not utilize additional positional embeddings.

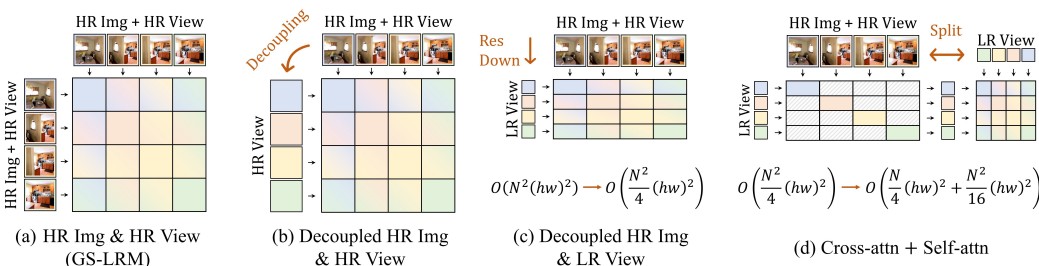

Figure 2: The proposed scalable architectural designs by decoupling viewpoint and image tokens, and modeling the global interactions via cross- and self-attentions ($N$: # views, $h = H/p, w = W/p$).

**Multi-view image tokenization.** For each input view image, which provides visual guidance to the reconstruction process, we extract both image features and corresponding pose information. Specifically, we divide an input image into non-overlapping patches and obtain two sets: RGB image patches and Plücker ray patches. These are then concatenated and linearly projected to construct the image patch tokens, $S_i \in \mathbb{R}^{HW/p^2 \times d}$,

$$S_{ij} = \text{Linear}(\text{concat}(I_{ij}, P_{ij})) \in \mathbb{R}^d, \qquad (2)$$

where $I_{ij} \in \mathbb{R}^{3p^2}, P_{ij} \in \mathbb{R}^{6p^2}$ represent the flattened $j$-th RGB image and Plücker ray patches for the $i$-th view, respectively, and $HW/p^2$ is the number of tokens for each input view image.

**Scalable multi-view context modeling.** Fig. 2-(a) shows the typical feed-forward 3D methods (Zhang et al., 2025; Chen et al., 2025; Xu et al., 2025) using transformer architecture, which perform full attention across multi-view images, incurring a quadratic increase in computational cost with respect to both the number of views and the image resolution. Fig. 2-(b) depicts our decoupling approach. Thanks to the decoupling technique, we can reduce the viewpoint resolution while still leveraging high-resolution multi-view images (Fig. 2-(c)). We further decrease the computation cost by two-stage multi-view context modeling, per-view cross-attention and viewpoint self-attention (Fig. 2-(d)). For example, given 16 input images with a resolution of $256 \times 256$ and a patch size of 8, the relative computational cost follows the ratio (1:1:0.25:0.08, Fig. 2-(a):(b):(c):(d)), highlighting that our proposed method can accommodate more input views with significantly less computational burden.

**Update block.** Given a set of viewpoint tokens, we formulate the problem as an iterative refinement process, where the viewpoint tokens are progressively updated through interactions with multi-view image tokens, ultimately leading to more accurate and spatially consistent 3D Gaussian Splatting. As shown in Fig. 3, our model consists of multiple transformer modules, each comprising one cross-attention layer followed by one self-attention layer.

$$\tilde{V}_i^{(l-1)} = \text{cross-attn}^{(l)}(V_i^{(l-1)}, S_i), \quad (3)$$

$$\{V_i^{(l)}\}_{i=1}^N = \text{self-attn}^{(l)}(\{\tilde{V}_i^{(l-1)}\}_{i=1}^N), \quad (4)$$

where the superscript $(l)$ denotes the layer index. In the cross-attention layers, the viewpoint tokens are refined by the visual information from their corresponding image tokens. In the self-attention layers, the viewpoint tokens interact with each other to refine and enhance their representations by incorporating global contextual information. Note that we use separate model parameters for the update blocks at different layers.

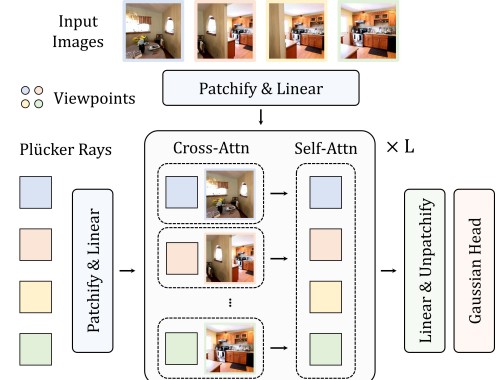

Figure 3: Network architecture.

**Token uplifting.** Standard cross-attention operations are typically applied between token sets with similar spatial resolutions, which allows for a straightforward one-to-one correspondence between queries and keys. In our setting, however, the cross-attention between viewpoint and image tokens

involves mismatched resolutions: we intentionally use lower-resolution (LR) viewpoint tokens compared to image tokens to improve architectural scalability and enhance the efficiency of 3D Gaussian generation.

This reduction in viewpoint token resolution, while beneficial for scalability and efficiency, may constrain their ability to fully capture and integrate the rich information present in the high-resolution (HR) image tokens. To bridge this gap, we propose a token uplifting strategy that aligns the LR viewpoint tokens with the HR image tokens. Specifically, each LR viewpoint token is lifted by a linear query layer that increases its feature dimension by a factor of $k$, resulting in the viewpoint token tensor of shape $H^v W^v / p^2 \times dk$. Then, it is reshaped to expand the token to the shape of $H^v W^v k / p^2 \times d$, where each original token now corresponds to $k$ finer-grained query tokens, allowing for better capture of visual correspondence during attention against image tokens.

After performing cross-attention with HR image tokens serving as keys and values, the updated tokens tensor of shape $H^v W^v k / p^2 \times d$ are reshaped back to the original viewpoint tensor of shape $H^v W^v / p^2 \times dk$, and subsequently projected back to the original embedding dimension via a linear projection layer, resulting in the refined viewpoint token tensor of shape $H^v W^v / p^2 \times d$. This technique preserves the updated information while maintaining computational efficiency in subsequent self-attention layers. Balancing representational capacity and efficiency, we empirically set the expansion factor to $k = 2$.

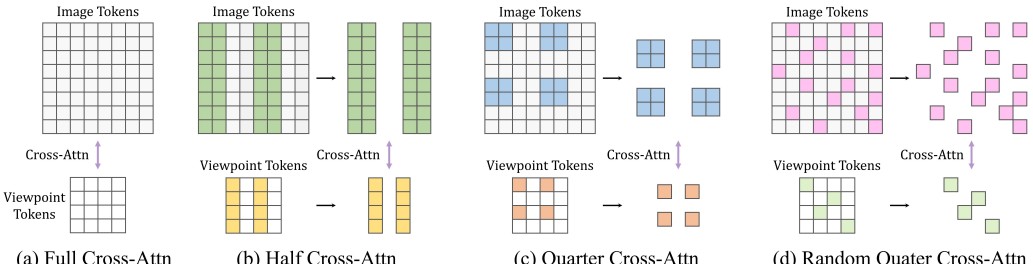

(a) Full Cross-Attn        (b) Half Cross-Attn        (c) Quarter Cross-Attn        (d) Random Quater Cross-Attn

Figure 4: Various mini-batch cross-attention schemes.

**Mini-batch cross-attention.** In our architecture, viewpoint tokens are iteratively updated at each layer based on information from the image tokens via cross-attention. The proposed decoupled viewpoint token design allows us to arbitrarily reduce the number of viewpoint tokens for improved scalability, whereas the resolution of image tokens remains fixed due to their spatial nature. Consequently, the primary computational bottleneck in cross-attention lies in the high-resolution image tokens.

To address this, we propose several efficient cross-attention schemes, as illustrated in Fig. 4, aimed at improving scalability without sacrificing performance. Our design is conceptually inspired by mini-batch gradient descent in optimization, where only a subset of data points is sampled in each iteration to reduce computational cost. Similarly, our mechanism selectively samples subsets of both image tokens and viewpoint tokens during cross-attention. While random token sampling (Fig. 4-(d)) is ideal in theory, it complicates efficient implementation. To mitigate this, we design structured sampling strategies that are simple to implement and demonstrate strong empirical performance.

**Decoding into 3D Gaussians.** After the final self-attention layer, we decode the final layer's viewpoint tokens, $V_i^{(L)}$, into Gaussian parameters through a single linear layer and apply post-activation functions. For a detailed description, please refer to our supplementary materials.

### 2.3 Training Objectives

After generating 3D Gaussians from viewpoint tokens, we rasterize them to the target viewpoint to obtain rendered images, $\hat{I}_t$. These rendered images are then supervised using ground-truth images, $I_t$, through MSE loss and perceptual loss (Chen & Koltun, 2017; Li et al., 2020). For each training scene, our training loss function is as follows.

$$\mathcal{L}_{\text{total}} = \sum_{t \in \mathcal{T}} \mathcal{L}_{\text{MSE}}(\hat{I}_t, I_t) + \lambda \mathcal{L}_{\text{perceptual}}(\hat{I}_t, I_t), \tag{5}$$

where $\mathcal{T}$ is a set of target view indices, $\lambda$ is a weighting factor that balances the contribution of the perceptual loss relative to the MSE loss, which we set to 0.5.

## 3 EXPERIMENTS

### 3.1 DATASETS

We train our model on two large-scale datasets: RealEstate10K (RE10K) (Zhou et al., 2018), DL3DV (Ling et al., 2024), and evaluate it on three datasets, including ACID (Liu et al., 2021), which is used only for evaluation. We adopt the RE10K split following prior work (Charatan et al., 2024) and use the official split for DL3DV. We use an image resolution of $256 \times 256$ for the RE10K and ACID datasets, while for the DL3DV dataset, we use a resolution of $256 \times 448$ and $512 \times 960$. In addition, we employ the undistorted version of the DL3DV dataset at a resolution of $540 \times 960$, which originates from LongLRM (Ziwen et al., 2025).

### 3.2 IMPLEMENTATION AND TRAINING DETAILS

Our model consists of 12 update layers, each containing one cross-attention and one self-attention block. Inside each attention module, we adopt a pre-normalization method with LayerNorm (Ba et al., 2016) and QK-Norm (Henry et al., 2020) method with an RMSNorm (Zhang & Sennrich, 2019) layer. Also, each of them utilizes multi-head attention (Vaswani et al., 2017) with 12 heads and two linear layers with GELU (Hendrycks & Gimpel, 2016) activation. We set the hidden dimension of every linear layer to $d = 768$, and use a patch size of $p = 8$.

To enhance training efficiency, we utilize FlashAttention-2 (Dao, 2023) to improve attention computation efficiency, apply gradient checkpointing (Chen et al., 2016) to reduce memory overhead, and adopt mixed-precision training with BFloat16 to accelerate computations while maintaining numerical stability. For more details, please refer to the supplementary material.

### 3.3 EVALUATION

We compare our model against recent generalizable 3D reconstruction methods (Charatan et al., 2024; Chen et al., 2025; Zhang et al., 2025; Xu et al., 2025; Nam et al., 2025; Ziwen et al., 2025) as well as optimization-based approaches (Kerbl et al., 2023; Yu et al., 2024). For evaluation, we follow the evaluation settings from (Charatan et al., 2024; Ye et al., 2024) for RE10K and (Xu et al., 2025; Ziwen et al., 2025) for DL3DV. We denote our various viewpoint settings as $(V, H/F, F)$, where $V$ is the number of viewpoints, and the following entries indicate the resolutions of viewpoint and image tokens ($H$: half-resolution, $F$: full-resolution). For example, a setting of $(2, H, F)$ indicates two viewpoints with half-resolution viewpoint tokens and full-resolution image tokens. $MC$ refers to our quarter mini-batch cross-attention (Fig. 4-(c)). Note that our 2-view full-resolution setting $(2, F, F)$ does not include token uplifting, as the viewpoint and image resolutions are identical. Additionally, when utilizing more viewpoints than the evaluation protocol, we sample extra viewpoints between the input viewpoints, ensuring that these samples do not overlap with the target indices.

### 3.4 RESULTS

In Tab. 1, 2 and Fig. 5, we compare our approach with feed-forward methods on the RE10K dataset and cross-dataset generalization on ACID and DL3DV, which serve as the standard in related works. Furthermore, we report results with an increased number of input views (4 and 8), which incur less than half of the computation time compared to the baseline (DepthSplat) while achieving superior performance. For the DL3DV dataset, our method consistently outperforms the baselines across various viewpoint and resolution configurations, including inference speed and memory usage, while achieving efficient scene representation with fewer Gaussians, as shown in Tab. 3, 4 and Fig. 6, 7. While DepthSplat and our method are both trained under varying numbers of input views, our approach demonstrates enhanced scalability with respect to the number of views in Tab. 3.

In the wide-coverage setting (Tab. 5), following the LongLRM protocol (Ziwen et al., 2025), we evaluate using 32 high-resolution input images under full-frame coverage. For comparison, we also include optimization-based methods such as 3D-GS (Kerbl et al., 2023) and Mip-Splatting (Yu et al., 2024), both trained for 30k iterations using only the input images. LongLRM$_{10}$ means finetuning 10 epochs initialized from the LongLRM's generated Gaussians. Since our approach produces more compact 3D Gaussian representations, the finetuning process is significantly faster than the baseline.

### 3.5 COMPUTATIONAL COSTS OF TRAINING

We report detailed comparisons of computational costs during training in Tab. 6. The iteration time is measured under the same setting: half-resolution 8 viewpoints $(8, H, F)$, and a batch size of 16

| Method | #Param (M) | PSNR ↑ | SSIM ↑ | LPIPS ↓ | # of Gaussians | Time (S) |
|---|---|---|---|---|---|---|
| pixelSplat (Charatan et al., 2024) | 125 | 25.89 | 0.858 | 0.142 | 131,072 | 0.101 |
| MVSplat (Chen et al., 2025) | 12 | 26.39 | 0.869 | 0.128 | 131,072 | 0.047 |
| GS-LRM* (Zhang et al., 2025) | 300 | 28.10 | 0.892 | 0.114 | 131,072 | - |
| DepthSplat (Xu et al., 2025) | 354 | 27.47 | 0.889 | 0.114 | 131,072 | 0.065 |
| Gen-Den Nam et al. (2025) | 28 | 27.08 | 0.879 | 0.120 | 347,072 | 0.224 |
| Ours $(2, F, F)$ | 171 | **28.65** | **0.900** | **0.110** | 131,072 | **0.025** |
| Ours $(4, H, F)$ | 185 | 30.37 | 0.923 | 0.095 | 65,536 | 0.027 |
| Ours-MC $(4, H, F)$ | 185 | 30.10 | 0.919 | 0.098 | 65,536 | 0.027 |
| Ours $(8, H, F)$ | 185 | 31.57 | 0.935 | 0.082 | 131,072 | 0.028 |
| Ours-MC $(8, H, F)$ | 185 | 30.96 | 0.931 | 0.088 | 131,072 | 0.030 |

Table 1: Quantitative comparisons on the RE10K dataset with various viewpoint configurations. Inference time is the reconstruction time (not rendering speed), using a single NVIDIA RTX 4090 GPU. * indicates methods for which the official code and the pretrained models are not available.

| Method | ACID (Liu et al., 2021) | | | DL3DV (Ling et al., 2024) | | |
|---|---|---|---|---|---|---|
| | PSNR ↑ | SSIM ↑ | LPIPS ↓ | PSNR ↑ | SSIM ↑ | LPIPS ↓ |
| MVSplat (Chen et al., 2025) | 28.15 | 0.841 | 0.147 | 22.65 | 0.737 | 0.191 |
| DepthSplat (Xu et al., 2025) | 28.37 | 0.847 | **0.141** | 24.28 | 0.813 | 0.147 |
| Gen-Den (Nam et al., 2025) | 28.61 | 0.847 | **0.141** | 22.92 | 0.750 | 0.188 |
| Ours $(2, F, F)$ | **29.24** | **0.856** | 0.143 | **25.35** | **0.826** | **0.144** |
| Ours $(4, H, F)$ | 30.10 | 0.877 | 0.138 | 27.90 | 0.877 | 0.122 |
| Ours-MC $(4, H, F)$ | 29.90 | 0.873 | 0.141 | 27.68 | 0.881 | 0.127 |
| Ours $(8, H, F)$ | 30.96 | 0.894 | 0.122 | 29.56 | 0.907 | 0.101 |
| Ours-MC $(8, H, F)$ | 30.47 | 0.887 | 0.132 | 29.03 | 0.900 | 0.110 |

Table 2: Cross-dataset generalization results on the ACID and DL3DV datasets (256×256). All models are trained on the RE10K training set and evaluated on each dataset.

| Method (50-frame baseline) | Views | PSNR ↑ | SSIM ↑ | LPIPS ↓ | # of Gaussians | Time (S) | Memory (GB) |
|---|---|---|---|---|---|---|---|
| MVSplat (Chen et al., 2025) | 6 | 22.93 | 0.775 | 0.193 | 688,128 | 0.279 | 5.87 |
| DepthSplat (Xu et al., 2025) | 6 | 24.19 | 0.823 | **0.147** | 688,128 | 0.102 | 3.55 |
| | 11 | 24.28 | 0.833 | 0.141 | 1,261,568 | 0.170 | 6.01 |
| | 24 | 22.37 | 0.781 | 0.195 | 2,752,512 | 0.371 | 12.39 |
| Ours | $(6, H, F)$ | **25.60** | **0.830** | 0.168 | **172,032** | **0.031** | **1.40** |
| | $(11, H, F)$ | **26.99** | **0.865** | **0.140** | **315,392** | **0.051** | **1.59** |
| | $(24, H, F)$ | **27.38** | **0.882** | **0.126** | 688,128 | **0.123** | **2.01** |

Table 3: Quantitative comparisons on the DL3DV dataset with various view configurations (256×448). Inference time and memory consumption are measured only during the Gaussian generation stage, excluding the rendering process on a single NVIDIA RTX 4090 GPU.

| Method (100-frame baseline) | Views | PSNR ↑ | SSIM ↑ | LPIPS ↓ | # of Gaussians | Time (S) | Memory (GB) |
|---|---|---|---|---|---|---|---|
| DepthSplat (Xu et al., 2025) | 12 | 21.38 | 0.739 | 0.265 | 5,898,240 | - | OOM |
| Ours | $(12, H, F)$ | **24.35** | **0.781** | **0.256** | **1,474,560** | **0.415** | **3.53** |

Table 4: Quantitative comparisons on the DL3DV dataset on high-resolution setting (512×960). Inference time and memory consumption are measured only during the Gaussian generation stage, excluding the rendering process on a single NVIDIA RTX 4090 GPU. Since DepthSplat encounters out-of-memory issue on the device, we evaluate its performance using a single H100 GPU.

on a single RTX 4090 GPU. For memory comparison, to provide a clearer analysis, all models are run without gradient checkpointing on a single H100 GPU. Lastly, we present a theoretical comparison of FLOPs that further underscores the efficiency of our method, with only a marginal drop in performance. For detailed calculation, please refer to our supplementary material.

### 3.6 ABLATIONS AND ANALYSIS

Tab. 7 presents the ablations on the number of layers. All variants are trained under half-resolution 4 viewpoints setting $(4, H, F)$, with a batch size of 16 on a single RTX 4090 GPU. The results demonstrate the scalability of our transformer-based architecture, showing consistent performance

| Method (full baseline; average 350-frame) | views | Time ↓ | PSNR ↑ | SSIM ↑ | LPIPS ↓ |
|---|---|---|---|---|---|
| 3D-GS (30k) (Kerbl et al., 2023) | 32 | 13min (A100) | 23.60 | 0.779 | 0.213 |
| Mip-splatting (30k) (Yu et al., 2024) | 32 | 13min (A100) | 23.32 | 0.784 | 0.217 |
| LongLRM (Ziwen et al., 2025) | 32 | 0.61sec (H100) | 24.10 | 0.783 | **0.254** |
| Ours | $(32, H, F)$ | **0.53sec** (H100) | **24.30** | **0.803** | 0.257 |
| LongLRM$_{10}$ | 32 | 37sec (A100) | 25.60 | 0.826 | 0.233 |
| Ours$_{10}$ | $(32, H, F)$ | **7sec** (A100) | 25.67 | 0.844 | 0.230 |
| Ours$_{50}$ | $(32, H, F)$ | 34sec (A100) | **26.46** | **0.862** | **0.210** |

Table 5: Quantitative comparisons on the undistorted and wide-coverage DL3DV dataset (540×960). We utilized flash attention v3 (Shah et al., 2024) when zero-shot inference using a single H100 GPU.

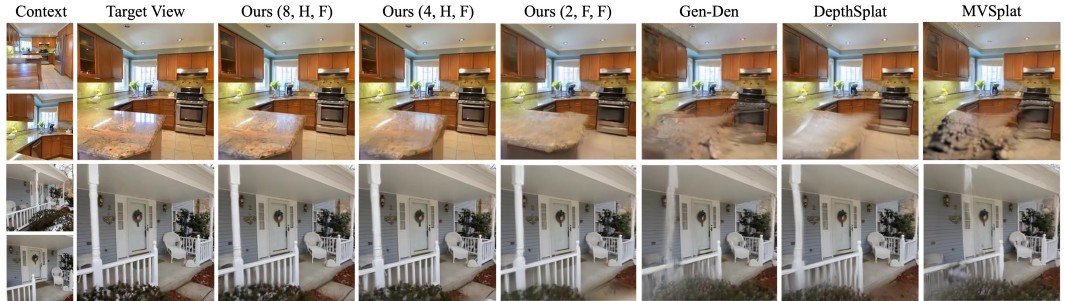

Figure 5: Qualitative comparison of novel view synthesis on the RE10K dataset (256×256).

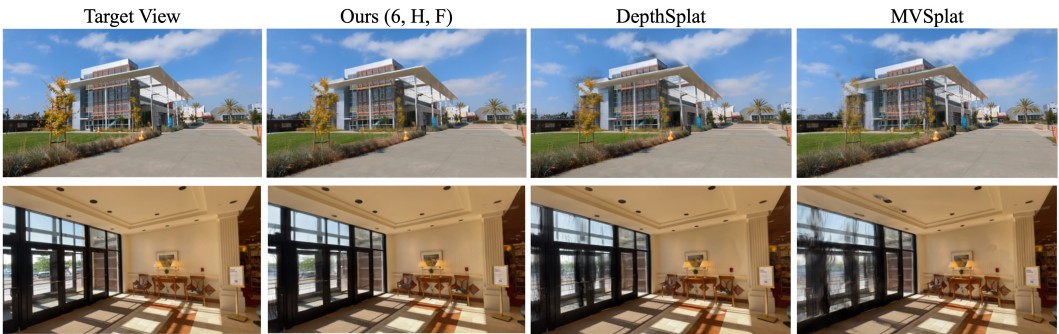

Figure 6: Qualitative comparison of novel view synthesis on the DL3DV dataset (256×448).

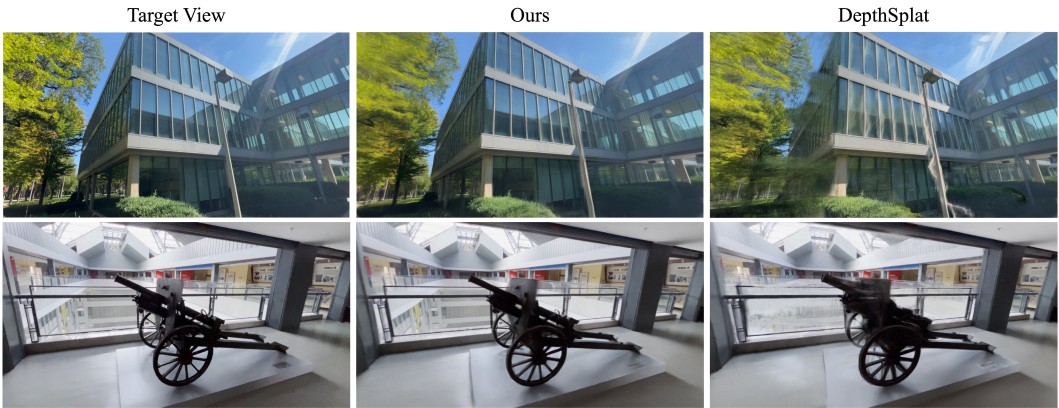

Figure 7: Qualitative comparison of novel view synthesis on the DL3DV dataset (512×960).

gains as the number of layers increases. Also, from the perspective of the iterative refinement procedure, increasing the number of layers can be interpreted as introducing more optimization steps, which aligns with our intuition that deeper refinement leads to more accurate 3D representations.

| Method | PSNR ↑ | SSIM ↑ | LPIPS ↓ | Iteration time (S) | Training Memory (GB) | GFLOPs |
|---|---|---|---|---|---|---|
| Baseline | **30.39** | **0.923** | **0.095** | 1.51 | 62.5 | 3.83 |
| w/ Half Cross-attn | 30.25 | 0.922 | 0.096 | 1.13 | 47.4 | 1.71 |
| w/ Quater Cross-attn | 30.08 | 0.919 | 0.098 | **0.94** | **39.0** | **0.81** |

Table 6: Quantitative comparisons of our mini-batch cross-attention on the RE10K dataset.

| | # Params | PSNR ↑ | SSIM ↑ | LPIPS ↓ |
|---|---|---|---|---|
| 12 layers (base) | 185M | **29.24** | **0.907** | **0.109** |
| 9 layers | 139M | 29.01 | 0.903 | 0.112 |
| 6 layers | 94M | 28.68 | 0.898 | 0.116 |
| 3 layers | 48M | 28.04 | 0.887 | 0.126 |

| | PSNR ↑ | SSIM ↑ | LPIPS ↓ |
|---|---|---|---|
| Baseline | **29.24** | **0.907** | **0.109** |
| w/o iter. refinement | 28.58 | 0.893 | 0.127 |
| w/o resolution decoupling | 28.47 | 0.891 | 0.123 |
| w/o token uplifting | 28.90 | 0.901 | 0.113 |

Table 7: Ablations on model size.      Table 8: Ablations on model architecture.

In Tab. 8, we present the ablation results on key architectural components to validate the effectiveness of each design choice. All experiments are conducted under the same configuration as the model size ablation with 12 layers baseline. Further ablation studies are provided in the supplementary material.

**1) Iterative refinement.** The cross-attention blocks in our model keep providing visual (image) into the viewpoint tokens as part of the iterative refinement process. To validate our design choice, we conduct an ablation study in which only the first layer uses cross-attention, while the remaining layers use self-attention. The baseline consists of 12 layers, each composed of a cross-attention followed by a self-attention block, whereas the variant uses a single cross-attention layer followed by 23 self-attention layers. The table below shows that our consecutive cross-attention with image features plays a critical role in refining the viewpoint embeddings especially comparing LPIPS metric.

**2) Resolution decoupling.** We perform an ablation study where low-resolution images (matched to the viewpoint resolution) are used as visual inputs for the cross-attention layers. Compared to the baseline, which uses high-resolution images, this variant shows a performance drop, confirming the effectiveness of our decoupling strategy. It enables the use of high-resolution features while maintaining a compact 3D representation.

**3) Token uplifting.** Removing the token uplifting mechanism leads to a drop in performance across all metrics compared to baseline. This validates the importance of expanding low-resolution view tokens before cross-attention with high-resolution image tokens. Without this step, the model struggles to capture fine-grained spatial correspondences, resulting a degraded reconstruction quality.

## 4 LIMITATIONS

One limitation of this work is the computational bottleneck associated with self-attention across multiple input views. Although we significantly reduce the computational cost by utilizing compact viewpoint embeddings, challenges may arise as the number of input views increases considerably. In this study, aiming for scalable feed-forward 3D models, we present the first implementation of the proposed framework, a feed-forward 3D model that iteratively refines 3D representations by leveraging high-resolution image information at every layer. Further development of more scalable alternatives to self-attention would be a valuable direction for future research.

Additionally, our model requires known camera poses, typically obtained from structure-from-motion (SfM) (Schonberger & Frahm, 2016; Pan et al., 2024) methods such as COLMAP or from precisely synthesized datasets. However, these poses can be error-prone and are often difficult to acquire at scale. To enhance the practicality of our model and enable the use of giant-scale raw video datasets (Chen et al., 2024b), extending it to operate in a pose-free setting (Ye et al., 2024; Hong et al., 2024; Kang et al., 2025) remains an important and open direction for future work.

## 5 CONCLUSION

In this work, we present an iterative Large 3D Reconstruction Model (*iLRM*), a feed-forward architecture that reflects per-scene optimization-based schemes, by stacking multiple update layers composed of cross- and self-attention modules. By decoupling Gaussian representations from input images and splitting the update mechanism into per-view interactions with image features and global context aggregation over compact viewpoint embeddings, *iLRM* enables efficient, scalable, and high-quality 3D reconstruction across diverse scenes. We believe that *iLRM* lays a strong foundation for future research in feed-forward 3D reconstruction.

## ETHICS STATEMENT

This work is conducted with a strong commitment to ethical research practices. All experiments are carried out using publicly available datasets that have been released in compliance with established ethical and legal standards. Our study does not involve human subjects, personal data, or any sensitive information, and therefore does not raise concerns related to privacy, fairness, or potential harm. We are dedicated to maintaining the highest level of ethical integrity throughout both the development and dissemination of our work.

## REPRODUCIBILITY STATEMENT

We place a strong emphasis on the reproducibility of our research. All datasets, implementation details, and experimental protocols are carefully documented in the paper. Upon publication, we will release the full codebase in detail, ensuring that the community can replicate our findings and build upon them in future work.

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

## A    RELATED WORKS

### A.1    FEED-FORWARD 3D GAUSSIAN SPLATTING

Feed-forward 3D Gaussian Splatting (Charatan et al., 2024; Chen et al., 2025; Xu et al., 2025; Nam et al., 2025; Zhang et al., 2025; Szymanowicz et al., 2024b; Tang et al., 2025; Xu et al., 2024b) capitalizes on robust priors learned from extensive datasets to estimate Gaussian primitive parameters and synthesize novel view images using sparse input data. PixelSplat (Charatan et al., 2024) and LatentSplat (Wewer et al., 2024) predict Gaussians from image features using an epipolar line sampling method to enhance geometric accuracy, while MVSplat (Chen et al., 2025) and MVSGaussian (Liu et al., 2024) construct cost volumes through a plane-sweep stereo approach. In a further development, Flash3D (Szymanowicz et al., 2024a) and DepthSplat (Xu et al., 2025) introduce a pre-trained depth estimation model (Piccinelli et al., 2024; Yang et al., 2024), which improves the robustness of the spatial positions of 3D Gaussians. In contrast, GS-LRM (Zhang et al., 2025) and Long-LRM (Ziwen et al., 2025) minimize reliance on explicit 3D priors by leveraging large-scale data-driven priors.

While demonstrating strong results, a major limitation of all the aforementioned approaches lies in their non-scalable architectural design, which restricts their ability to effectively leverage a large number of input views. Moreover, the one-shot generation strategy, which produces 3D representations in a single forward pass, often fails to capture complex geometric details and fine 3D consistency, making them suboptimal for high-quality 3D reconstruction. We address these limitations by proposing an iterative 3D reconstruction framework and scalable architectural designs.

**Iterative refinements.** Our work is also closely related to recent methods that adopt iterative refinement strategies, such as G3R (Chen et al., 2024e) and Gen-Den (Nam et al., 2025). Both utilize actual gradients to update their representations more precisely. While promising, these approaches require additional computational burden for rendering multiple images per training iteration, and relying solely on gradients may risk overlooking valuable information present in the raw input images. Nonetheless, exploring how to incorporate gradient information remains an interesting direction for future work.

### A.2    3D REPRESENTATIONS FROM EMBEDDINGS

Inspired by previous generative approaches (Goodfellow et al., 2014; Karras et al., 2019; Chan et al., 2022), recent works (Hong et al., 2023; Chen et al., 2024a; Flynn et al., 2024) have investigated the synthesis of 3D representations directly from learnable embeddings, guided by input image supervision. This paradigm leverages the expressive capacity of latent spaces to encode rich geometric priors, which act as structural templates that guide the reconstruction process. Such approaches offer notable flexibility, allowing rendering from arbitrary viewpoints and adaptation to varying space scales and camera poses. However, both LRM (Hong et al., 2023) and Lara (Chen et al., 2024a) are limited to object-centric representations, restricting scalability to complex scenes involving multiple objects or large spatial layouts. The recently proposed Quark (Flynn et al., 2024) also utilizes learnable embeddings to fuse visual cues from multiple images, demonstrating compelling results, but its representation is confined to the target view (Xu et al., 2024a; Liu et al., 2024; Jin et al., 2024), lacking an explicit and persistent 3D reconstruction.

In contrast to previous works, we construct scene-level explicit 3D representations from viewpoint embeddings by decoupling the generation of Gaussians from the input images. This separation enables iterative refinement of the embeddings using low-level visual features and provides flexible control over the density of the 3D representation, independent of the input image resolution.

## B    ADDITIONAL IMPLEMENTATION DETAILS

We initialize model weights using a zero-mean normal distribution with a standard deviation of 0.02. Bias terms are omitted in all Linear and normalization layers. The model is trained using the AdamW (Loshchilov & Hutter, 2017) optimizer with hyperparameters $\beta_1 = 0.9$ and $\beta_2 = 0.095$. A weight decay of 0.05 is applied to all parameters except the weights of LayerNorm (Ba et al., 2016). We use a cosine learning rate schedule with a peak learning rate of 2e-4 and a warmup of

2500 iterations. Our training setup largely follows the configuration proposed in (Zhang et al., 2025; Jin et al., 2024).

For the RealEstate10K (RE10K) (Zhou et al., 2018) dataset, the 8-view half-resolution viewpoint setting *(8, H, F)* is trained on 8 H100 GPUs with a total batch size of 256 for 50,000 iterations. Similarly, the 4-view half-resolution viewpoint setting *(4, H, F)* is trained on 8 RTX 4090 GPUs with a total batch size of 128 for 100,000 iterations. The mini-batch cross-attention variants were also trained with the equivalent computational budgets for each viewpoint setting. Lastly, the 2-view full-resolution viewpoint setting *(2, F, F)*, which serves as the reference point, is trained on 8 H100 GPUs for 200,000 iterations.

There are two variants in the DL3DV dataset (Ling et al., 2024). **1)** For comparison with DepthSplat, we initialize from the pretrained *(8, H, F)* model trained on the RE10K dataset, and finetune it on 8 H100 GPUs with a total batch size of 96 for 100,000 iterations. During training, the number of input viewpoints is randomly sampled between 6 and 11 to expose the model to varying numbers of viewpoints/images (Xu et al., 2025). Following this stage, the model is further finetuned under the high-resolution setting (512×960). **2)** For comparison with LongLRM (Ziwen et al., 2025), which incorporates an undistortion preprocessing step, we adopt the training protocol described in the original work, using 8 H200 GPUs. The training resolution is scheduled in a curriculum of 256×256, 512×512, and 540×960.

**Gaussian representations.** After the final self-attention layer, the viewpoint features are decoded into Gaussian parameters using a single linear layer with an output dimension of 16. The Gaussian positions, denoted as $\mu$, consist of 5 channels: 2 for the spatial *xy* offset and 3 for depth, *z*. The final depth is obtained by averaging the 3 depth channels. Opacity ($\alpha$) is represented by a single channel. Covariance ($\Sigma$) is derived from 3 channels of scale and 4 channels of rotation. Finally, color (*c*) is represented using 3 channels. Higher-order spherical harmonics coefficients are not used in our method. The post-activation functions for each parameter follow the design of GS-LRM (Zhang et al., 2025), except for the spatial *xy* offset, for which we constrain the range to lie within a single pixel of viewpoint resolution. We utilize gsplat (Ye et al., 2025), an open-source library for Gaussian Splatting Kerbl et al. (2023) for a rasterizer.

**Camera pose normalization.** We normalize camera poses to align the scene into a consistent coordinate system and scale. First, we compute the average position and viewing directions (forward, down, and right) from the input camera extrinsics. These are used to build a new reference pose, which centers and aligns the scene. All camera extrinsics are then transformed into this reference frame. Finally, we scale the entire scene so that the largest camera distance is 1, ensuring the scene fits within a normalized space (Zhang et al., 2025).

## C ADDITIONAL ARCHITECTURAL DETAILS

We provide the detailed figure of our token uplifting module in Fig. 8. Note that, to balance the model's representational capacity and computational efficiency, the length of the low-resolution viewpoint embeddings does not exceed that of the high-resolution image features.

**Tokenization and normalization.** After tokenizing the viewpoints and multi-view images using linear layers, both types of tokens are passed through a LayerNorm (Ba et al., 2016). In each cross-attention layer, only the viewpoint tokens are further processed with a pre-normalization layer. Additionally, after the query and key linear projections, both tokens are passed through an extra normalization layer, referred to as the QK-Norm (Henry et al., 2020).

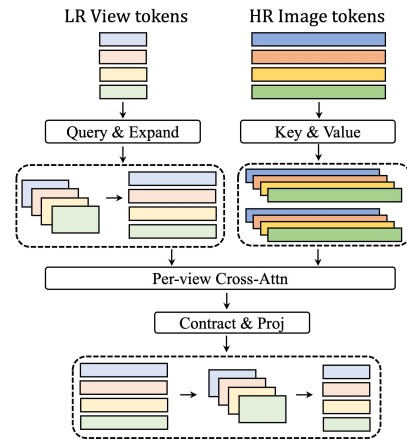

Figure 8: Token uplifting.

## D  ADDITIONAL EVALUATION DETAILS

When we utilize more input viewpoints (more than two in RealEstate10K (Zhou et al., 2018) experiment compared to the baselines), we sample additional viewpoints/images evenly between the two endpoint indices, ensuring that these samples do not overlap with the target indices. For cross-dataset generalization on the DL3DV dataset, we use a baseline of 12 frames.

In wide-baseline setting, every 8th image in the sequence is reserved for the test split, while K-means clustering on camera positions and viewing directions is applied to the remaining images to select input views that ensure wide scene coverage.

## E  ADDITION ABLATIONS ON MODEL ARCHITECTURE

We provide additional ablation studies and analyses in Tab. 9 under the same configuration as the ablations on model architecture in main script. All variants are trained under half-resolution 4 viewpoints setting $(4, H, F)$, with a batch size of 16 on a single RTX 4090 GPU.

|  | PSNR ↑ | SSIM ↑ | LPIPS ↓ |
|---|---|---|---|
| Baseline | **29.24** | **0.907** | **0.109** |
| w/o self-attention | 23.33 | 0.755 | 0.220 |
| w/ group-attention | 29.02 | 0.904 | 0.112 |
| w/ random init. | 28.90 | 0.902 | 0.112 |
| w/ LR-feature init. | 28.35 | 0.894 | 0.121 |

Table 9: Additional ablations.

**1) Self-attention.** To ensure a fair comparison, we replaced all self-attention layers with cross-attention layers rather than simply removing them, maintaining a comparable parameter count. The performance dropped significantly, highlighting the essential role of self-attention in capturing global dependencies and enhancing multi-view awareness among viewpoint embeddings. Without self-attention, the model struggles to integrate contextual information across different viewpoints, resulting in poor convergence and reconstruction quality.

**2) Group-attention** This variant replaces the per-viewpoint cross-attention mechanism with a group-attention approach, where all viewpoint tokens and image tokens are concatenated and jointly processed through a cross-attention block. Unlike our default design, group-attention introduces global interactions across all views. While this mechanism can increase the expressive capacity between multiple viewpoints, it incurs quadratic complexity with respect to the number of views. However, the increased computational cost does not yield performance gains, suggesting that separating the roles—using cross-attention for localized image-view interactions and self-attention for global refinement across viewpoints—leads to a more efficient and effective architecture, which is also validated as Alternating-Attention in VGGT (Wang et al., 2025a).

**3) Different initialization.** We also investigate the different initialization methods of scene representation. For the random initizliation, we used a learnable embedding initialized with zero mean and 0.02 standard deviation, whereas for the LR feature variant, we used features extracted from low-resolution images. In the PSNR training curve, the LR feature variant rises more quickly in the early stages but is later surpassed by the random initialization variant. We believe this is because, in our iterative cross-attention architecture, high-resolution image features and camera information are continually provided by the cross-attention blocks. As a result, the learnable embedding can offer more flexible and informative parameters for guiding iterative updates, whereas the LR image features may introduce redundant and less discriminative information that limits long-term performance gains. Moreover, the use of LR features may bias the early attention stages toward coarse processing, which can hinder the model's ability to fully refine fine details in later stages.

## F  COMPUTATIONAL COSTS OF TRAINING

We provide a detailed theoretical calculation of the FLOPs for our mini-batch cross-attention mechanism. In this analysis, we limit the computation to a per-view, single cross-attention operation, excluding our token uplifting strategy (as it introduces a constant cost across all variations). Given a viewpoint token of shape $(L_v, D)$ and an image token of shape $(L_i, D)$, where $L_v$ and $L_i$ denote the token lengths and $D$ is the hidden dimension, the FLOPs for the cross-attention operation are computed as:

$$4D^2(L_v + L_i) + 4L_v L_i D.$$

Assuming a hidden dimension of $D = 768$, an image resolution of $256 \times 256$, and a viewpoint resolution of $128 \times 128$, with a patch size of $8 \times 8$, the token lengths are computed as $L_i = 1024$ for the image tokens and $L_v = 256$ for the viewpoint tokens, based on the experimental configuration used in the RealEstate10K Zhou et al. (2018) dataset.

Thus, the computation becomes: baseline: **3.83 GFLOPs**; half cross-attention: **1.71 GFLOPs**; quarter cross-attention: **0.81 GFLOPs**.

# G    ADDITIONAL QUANTITATIVE RESULTS

**Varying baseline range.** We compare our model against recent generalizable 3D reconstruction methods (Chen et al., 2025; Xu et al., 2025; Nam et al., 2025) on the RealEstate10K (Zhou et al., 2018) dataset, with a particular focus on handling varying degrees of camera overlap (Ye et al., 2024). These overlap categories are determined using the dense feature matching method RoMA (Edstedt et al., 2024). As shown in Tab. 10, our method, which efficiently handles a large number of input viewpoints/images, achieves superior performance compared to existing approaches, especially in challenging cases with small viewpoint overlap (i.e., wide-baseline settings).

| Method | Small | | | Medium | | | Large | | | Average | | |
|---|---|---|---|---|---|---|---|---|---|---|---|---|
| | PSNR ↑ | SSIM ↑ | LPIPS ↓ | PSNR ↑ | SSIM ↑ | LPIPS ↓ | PSNR ↑ | SSIM ↑ | LPIPS ↓ | PSNR ↑ | SSIM ↑ | LPIPS ↓ |
| MVSplat (Chen et al., 2025) | 20.37 | 0.725 | 0.250 | 23.81 | 0.814 | 0.172 | 27.47 | 0.885 | 0.115 | 24.01 | 0.812 | 0.175 |
| DepthSplat (Xu et al., 2025) | 22.82 | 0.798 | 0.193 | 25.38 | 0.851 | 0.145 | 28.32 | 0.900 | 0.104 | 25.59 | 0.852 | 0.145 |
| Gen-Den (Nam et al., 2025) | 21.10 | 0.744 | 0.234 | 24.57 | 0.828 | 0.162 | 28.26 | 0.895 | 0.108 | 24.77 | 0.826 | 0.164 |
| Ours $(2, F, F)$ | **23.82** | **0.813** | **0.184** | **26.54** | **0.864** | **0.139** | **29.43** | **0.910** | **0.103** | **26.70** | **0.864** | **0.140** |
| Ours $(4, H, F)$ | 27.65 | 0.887 | 0.127 | 29.13 | 0.908 | 0.108 | 30.73 | 0.926 | 0.092 | 29.22 | 0.908 | 0.108 |
| Ours-MC $(4, H, F)$ | 27.41 | 0.882 | 0.131 | 28.87 | 0.904 | 0.111 | 30.44 | 0.927 | 0.095 | 28.96 | 0.904 | 0.112 |
| Ours $(8, H, F)$ | 29.44 | 0.912 | 0.106 | 30.51 | 0.925 | 0.093 | 31.77 | 0.937 | 0.080 | 30.61 | 0.925 | 0.092 |
| Ours-MC $(8, H, F)$ | 28.92 | 0.906 | 0.112 | 29.97 | 0.920 | 0.098 | 31.15 | 0.932 | 0.086 | 30.05 | 0.920 | 0.098 |

Table 10: Quantitative comparisons on the RE10K dataset under varying view overlap conditions.

**Same number of Gaussians.** We also validate the strength of our decoupling strategy in leveraging high-resolution images as visual cues while generating efficient and compact low-resolution 3D Gaussians. As discussed in our motivation, previous methods (Charatan et al., 2024; Chen et al., 2025; Zhang et al., 2025; Xu et al., 2025; Nam et al., 2025) require downsampling the input images to reduce the number of generated Gaussians, inherently coupling image resolution with representation density. To demonstrate the flexibility of our approach, we conduct an experiment in which all methods generate the same number of Gaussians using 4 viewpoints at half resolution. Specifically, the baseline methods (Chen et al., 2025; Xu et al., 2025) follow a $(4, H, H)$ configuration, where both the number of viewpoints and image resolution are reduced. In contrast, our method adopts a $(4, H, F)$ setting, where we preserve high-resolution image inputs while generating low-resolution Gaussians, thanks to our decoupled design. As shown in Tab. 11, our method surpasses the baselines in performance while requiring fewer computational resources in training, and faster inference speed, highlighting the practical advantages of our design. This result demonstrates the efficiency and the representational ability of our architecture, which effectively utilizes high-resolution visual cues, leading to superior reconstruction quality under the same output density without requiring expensive hardware. To train the baseline methods effectively with a large batch size (similar to ours), we run them on a single H100 GPU. Our method and MVSplat (Chen et al., 2025) are trained with a batch size of 16, while DepthSplat (Xu et al., 2025) is trained with a batch size of 12 due to memory constraints.

| Method | Params (M) | Train GPU (#) | PSNR ↑ | SSIM ↑ | LPIPS ↓ | # of Gaussians | Time (S) | Memory (GB) |
|---|---|---|---|---|---|---|---|---|
| MVSplat (Chen et al., 2025) | 12 | H100 (1) | 27.53 | 0.889 | 0.116 | 65,536 | 0.048 | **0.65** |
| DepthSplat (Xu et al., 2025) | 354 | H100 (1) | 28.08 | 0.898 | 0.107 | 65,536 | 0.062 | 2.49 |
| Ours | 185 | RTX 4090 (1) | 29.24 | 0.907 | 0.109 | 65,536 | **0.027** | 1.22 |
| Ours | 185 | RTX 4090 (2) | **29.82** | **0.916** | **0.101** | 65,536 | **0.027** | 1.22 |

Table 11: Quantitative comparisons under the same number of Gaussians on the RE10K dataset. Inference time and memory consumption are measured only during the Gaussian generation stage, excluding the rendering process on a single RTX 4090 GPU.

**Quarter resolution baselines.** To evaluate different viewpoint configurations—specifically the resolution of each viewpoint—we additionally compare a quarter-resolution variant of the viewpoint

inputs. All experiments in Tab. 12 are conducted using a single RTX 4090 GPU with a batch size of 16, 12 update layers, and trained for 100,000 iterations. While lowering the resolution of viewpoint inputs leads to a moderate drop in reconstruction quality, it significantly reduces the number of generated Gaussians, showing trade-off between accuracy and efficient representations.

| Method | PSNR ↑ | SSIM ↑ | LPIPS ↓ | # of Gaussians |
|---|---|---|---|---|
| Ours $(8, H, F)$ | **30.39** | **0.923** | **0.095** | 131,072 |
| Ours $(4, H, F)$ | 29.24 | 0.907 | 0.109 | 65,536 |
| Ours $(8, Q, F)$ | 27.36 | 0.868 | 0.152 | 32,768 |
| Ours $(4, Q, F)$ | 26.40 | 0.843 | 0.177 | **16,384** |

Table 12: Quantitative comparisons of different viewpoint configurations on the RE10K dataset. $Q$ denotes quarter resolution compared to the original image resolution.

## H  ADDITIONAL QUALITATIVE RESULTS

We present additional qualitative results in Fig. 9 and Fig. 10 corresponding to the RealEstate10K (RE10K) (Zhou et al., 2018) and DL3DV (Ling et al., 2024) datasets, respectively. Also, Fig. 11 presents high-resolution visualizations from the DL3DV dataset.

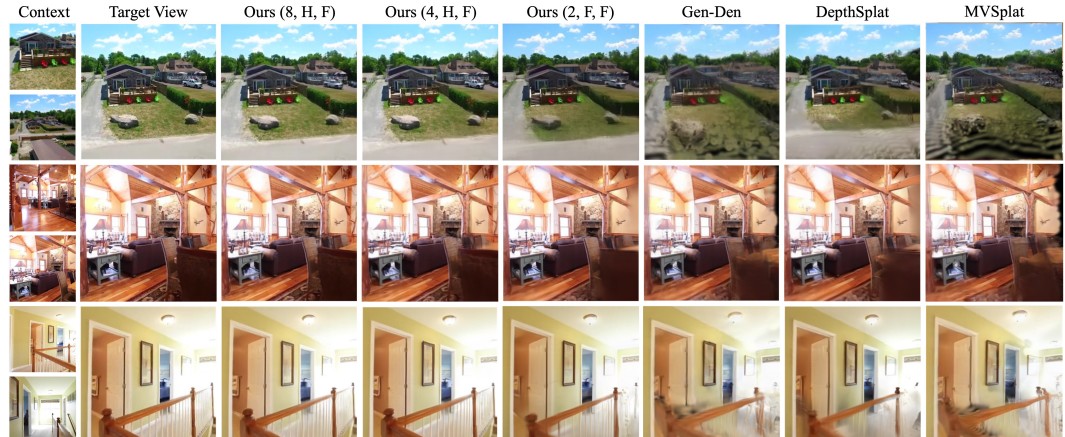

Figure 9: Qualitative comparison of novel view synthesis on the RE10K dataset (256×256).

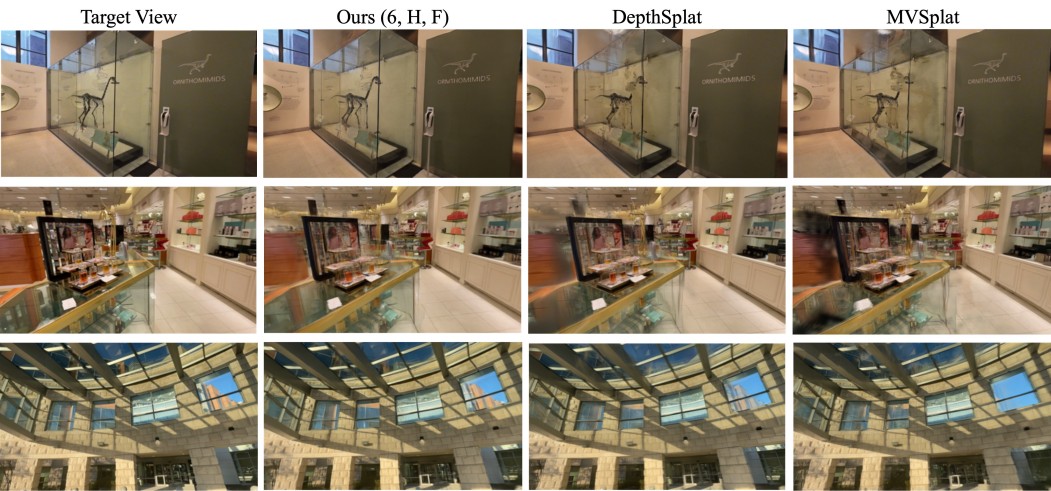

Figure 10: Qualitative comparison of novel view synthesis on the DL3DV dataset (256×448).

Target View                    Ours                    DepthSplat

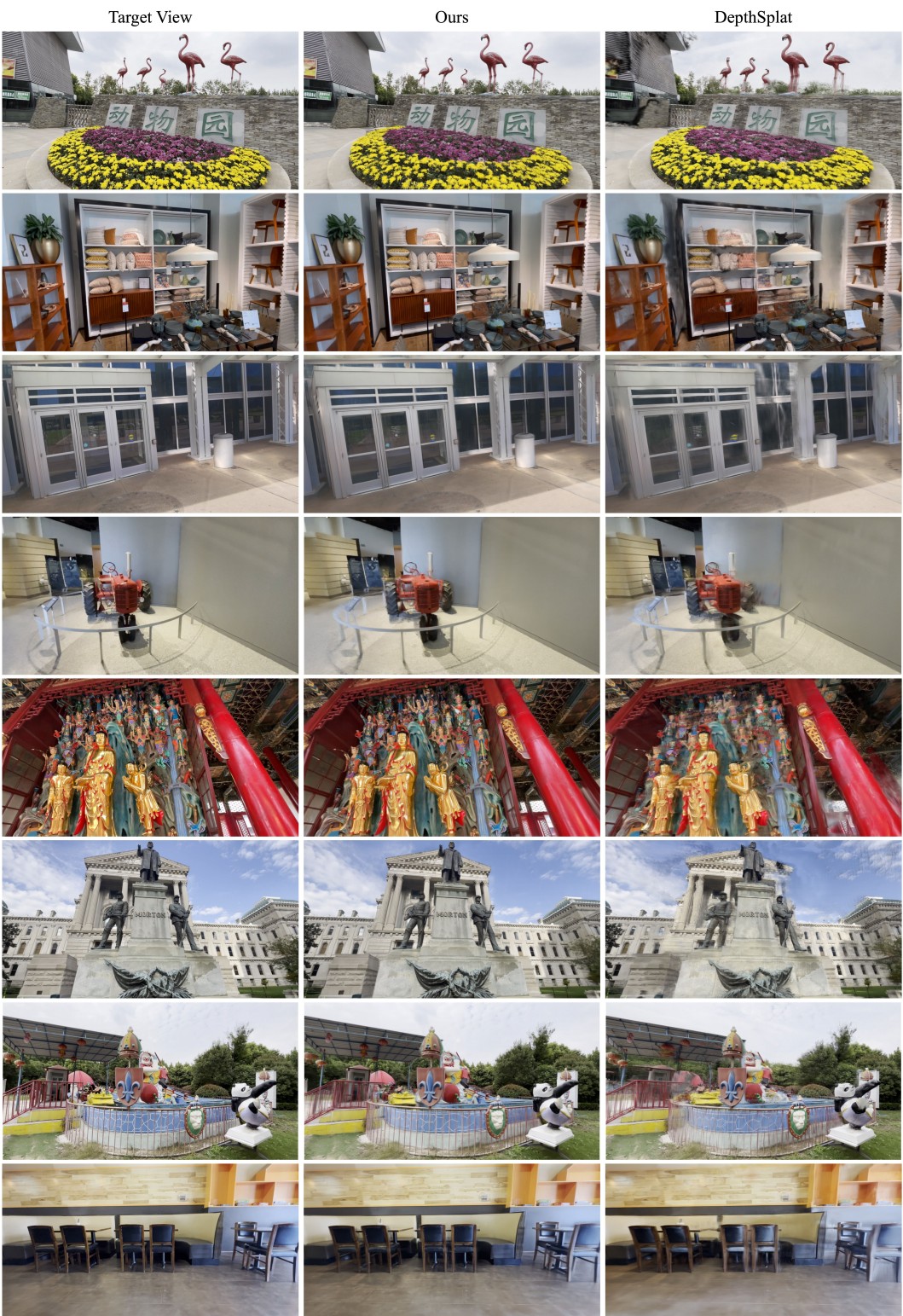

Figure 11: Qualitative comparison of novel view synthesis on the DL3DV dataset (512×960).

