# OpenReview forum: "iLRM: An Iterative Large 3D Reconstruction Model"
_ICLR.cc/2026/Conference — ICLR 2026 Conference Withdrawn Submission_

### Official Review · Reviewer_RdKH · 2025-10-25

**Soundness:** 2
**Presentation:** 3
**Contribution:** 2
**Rating:** 2
**Confidence:** 4

**Summary:**

his work proposes an iterative Large 3D Reconstruction Model, a feed-forward architecture that reflects per-scene optimization-based schemes. It introduces an efficient token update mechanism to enable iterative optimization of 3DGS and compression under dense views.

**Strengths:**

1. This work is well-written, and the description of the methodology section is clear.

2. It addresses an important problem, scalability issues in feed-forward 3DGS reconstruction. Compared to previous works, it effectively compresses the number of Gaussians under dense view inputs.

**Weaknesses:**

1. The term 'iterative' in the paper's title is difficult to understand. If I understand correctly, it is more similar to stacking attention blocks, following the scaling law, as shown in Table 7. Additionally, apart from Figure 1, the paper lacks more qualitative validation of 'iterative refinement.'

2. If I understand correctly, the core of iLRM lies in introducing view embeddings as tokens to be updated for reconstructing 3DGS (from Fig2(a) to Fig2(b)), thereby improving the efficiency of attention blocks and compressing the number of Gaussians. This does not bring new insights to the field and appears similar to the approach of LVSM [1], which is also used for feed-forward novel view synthesis tasks.

[1] LVSM: A Large View Synthesis Model with Minimal 3D Inductive Bias

3. The paper does not include a performance comparison with LVSM [1].

4. The paper lacks discussion on how to select the shape of the updated tokens tensor, which is important for iLRM. If they are manually set as hyperparameters, then performance may degrade for complex scenes that require more Gaussians.

5. The paper lacks a detailed description of the baseline in the ablation experiments. It is confusing that the baseline achieves better results. This section should discuss the performance of the initial framework without any of the proposed modules.

**Questions:**

1. Why does Figure 1 show that iLRM's performance improvement is significant as the number of layers increases, while Table 7 shows only marginal gains?

2. Why does iLRM appear to be faster with larger model parameters while having the same number of Gaussians in Table 1?

---

### Official Review · Reviewer_Twfa · 2025-10-29

**Soundness:** 2
**Presentation:** 3
**Contribution:** 2
**Rating:** 4
**Confidence:** 3

**Summary:**

The paper introduces iLRM, an iterative 3D reconstruction model that overcomes the severe scalability issues of previous feed-forward methods by decoupling the scene representation from the input images.

**Strengths:**

1. The model introduces an efficient two-stage attention mechanism that breaks the quadratic complexity bottleneck of prior methods. This allows it to effectively process a larger number of views and higher-resolution images without prohibitive computational costs.

2. iLRM reframes reconstruction as an iterative refinement process within a feed-forward network and achieves good reconstruction quality on standard benchmarks.

**Weaknesses:**

1. This paper only shows 2D novel view synthesis metrics like PSNR, SSIM, which are all about image quality. However, when it comes to reconstruction, the geometry is also very important. CD, F-score and similar metrics should be included.
2. No mesh reconstruction results. Showing conversion to a mesh would have better showcased the coherence of the underlying geometry and its practical applicability for downstream tasks like gaming or simulation.
3. Lack of comparison with feed-forward reconstruction methods like VGGT and its follow-ups.
4. Lack of novelty. Using cross-attention and similar methods to replace self-attention is a common technic for saving memory.

**Questions:**

See Weaknesses. More results about geometry reconstruction and comparison with VGGT should be given.

---

### Official Review · Reviewer_VqXS · 2025-10-29

**Soundness:** 3
**Presentation:** 2
**Contribution:** 2
**Rating:** 4
**Confidence:** 4

**Summary:**

This paper introduces iLRM, an optimized version of LRM mainly focusing on improving compute efficiency. iLRM utilizes many customization on original full attention layers from LRM to reduce its quadratic cost. iLRM also employs the concept of iterative refinement to guide their model design. The quality and efficiency improvements are effective as demonstrated in the evaluation results.

**Strengths:**

Decoupling representations and staged attention effectively tackle quadratic costs in multi-view processing, enabling more views (e.g., 8 vs. baselines' 2-4) with lower compute/memory. This new way of handling view-camera interaction could be helpful to reduce compute cost of general multi-view transformer models.

**Weaknesses:**

* While the concept of iterative refinement is nice and interesting, it is reluctant to say the current model design has a strong connection to the iterative refinement, especially the claimed “feedback-driven refinement” (L93). Since the LRM usually just stacks of attention block processing on the same series of tokens, one can also say that the tokens are “iteratively refined” block by block. The paper fails to convincingly show this decoupled representation enables unique iterative refinement.
* The proposed “token uplifting” is a trivial design for many cross-attention applications with unmatched channel dimensions of tokens.
* The mini-batch cross-attention is conceptually very similar to dropout operations. If so, simply introducing this part as a special dropout might be better. It is unnecessary to create such a new concept.

**Questions:**

None

---

### Note · Authors · 2025-11-12

I have read and agree with the venue's withdrawal policy on behalf of myself and my co-authors.